# When I Receive Too Much Social Support: The Effect of Social Support Overload on Users’ Life Burnout and Discontinuance in Fitness Apps

**DOI:** 10.3390/healthcare13020191

**Published:** 2025-01-19

**Authors:** Ruihan Li, Shuang Wang, Tailai Wu

**Affiliations:** School of Medicine and Health Management, Huazhong University of Science and Technology, Wuhan 430030, China; m202275736@hust.edu.cn (R.L.); m202275741@hust.edu.cn (S.W.)

**Keywords:** fitness apps, social support overload, basic psychological need frustration, discontinuance, life burnout

## Abstract

Background/Objectives: As fitness apps increasingly incorporate social interaction features, users may find themselves overwhelmed by an excess of received support, struggling to effectively manage it. Highlighting a novel recipient-centric perspective, we aim to investigate the impact of social support overload on users’ life burnout and discontinuance within fitness apps. Methods: Utilizing Social Support Theory and Basic Psychological Needs Theory, we develop a model to examine how emotional, network, and informational support overload affect life burnout and discontinuance through the frustration of basic psychological needs: autonomy, competence, and relatedness. A total of 443 fitness app users were included in our study, and we employed Structural Equation Modeling (SEM) to empirically test this model. Results: The results highlight the significant mediating role of the frustration of basic psychological needs between social support overload and life burnout/discontinuance. Network and informational support overload positively correlate with frustration of all needs, whereas emotional support overload shows a complex relationship. All need frustrations are linked to life burnout, but only autonomy and relatedness frustrations significantly lead to discontinuance. Additionally, gender and app use proficiency are significant control variables impacting discontinuance. Conclusions: This study adopts a novel recipient-centric perspective to explore social support overload, examining its effects on life burnout and discontinuance and offering practical implications for both users and app managers.

## 1. Introduction

Fitness apps, which are mobile applications that provide information and guidance on exercise, weight loss, and diet management, have significantly grown in popularity in recent years [1]. Serving as practical, low-cost solutions for fitness monitoring and self-health management, notable examples such as Strava, Nike+, Runtastic, and KEEP have achieved extensive user engagement. In January 2024, global downloads of the top fitness apps reached nearly 14 million [2]. The fitness app market is on a robust growth path, with a projected revenue of USD 6.86 billion in 2024 and an estimated compound annual growth rate of 7.96% from 2024 to 2029, culminating in a market volume of USD 10.06 billion by 2029 [3]. This upward trend was notably accelerated by the pandemic, as stay-at-home policies limited access to traditional exercise facilities. The convenience and flexibility of home workouts offered by fitness apps have firmly positioned them as a mainstay in the fitness landscape, with continued growth anticipated in the post-pandemic era.

Social support, a critical component embedded within the social interaction features of fitness apps, plays a vital role in their success. These features within the apps allow users to share fitness outcomes, seek fitness guidance, communicate with peers, and receive encouragement through “likes” or comments, facilitating both the receipt and provision of social support [4,5]. While such support can significantly boost users’ connection, engagement, and vitality, it also presents a potential downside: social support overload. This occurs when the level of social support exceeds a user’s capacity to process it effectively. Studies have highlighted that this negative social dimension of fitness apps can turn enjoyable activities into annoying virtual competitions and create social pressure, ultimately overwhelming users [6,7]. Unchecked, social support overload can cause user burnout and a decline in app engagement. Recognizing social support overload as a critical issue is vital due to its detrimental effects on users’ psychological well-being and behavioral outcomes, as well as its negative impact on fitness apps’ development. Therefore, to reduce the negative effects of social support overload, it is imperative to explore the mechanism underlying the effect of social support overload on users’ psychological and behavioral outcomes.

Previous research has predominantly explored social support overload within the context of social networking sites (SNSs), focusing on the perspective of social support providers (e.g., [8,9,10]). The term “social support overload” in this context typically refers to the exhaustion and feeling of being overwhelmed that users feel when they receive too many requests for social support and have to provide support on SNSs, mainly from a provider’s standpoint [11,12]. This overload has been linked to emotional exhaustion, role conflicts, and social media fatigue, prompting users to discontinue platform usage, predominantly in the context of SNSs [12,13,14]. As fitness apps increasingly incorporate social features, there remains a notable research gap concerning social support overload in these digital fitness environments, particularly from the recipient’s perspective. Furthermore, regarding the perspective of support recipients, existing research on social support overload in traditional in-person relationships—such as marriages [15,16], parent–child relationships [17], and employee–organization relationships [18]—does not capture the unique characteristics of the online and mobile contexts. Compared with the provider perspective, the recipient perspective provides a more precise depiction of the impact and consequences of social support overload, as it directly reflects individuals’ perceptions, interpretations, and experiences of the support that they receive. The recipient perspective allows for a more nuanced and empathetic exploration of the complexities of social support overload. Given the unexplored nature of social support overload in fitness apps and the scarcity of literature from the recipient perspective, it is crucial to investigate how this overload impacts users’ psychological and behavioral outcomes from the recipient perspective.

In the context of fitness apps, physical exercise has been proven to effectively alleviate individuals’ perception of life burnout, making it a relevant indicator of negative psychological outcomes. Additionally, post-adoption behavior, especially discontinuance, holds interest to both app developers and IS researchers. Accordingly, we employ life burnout as the psychological outcome—referring to a state of emotional and mental exhaustion [19]—and discontinuance as the behavioral outcome—reflecting users’ usage behavior of reducing their use of or abandoning fitness apps [20]. Therefore, our research question is proposed as follows:*How does social support overload influence users’ life burnout and discontinuance within fitness apps?*

To answer this question, we developed a research model based on the Social Support Theory and Basic Psychological Needs Theory. Data were collected from fitness app users via surveys to validate this research model, and the results of the analysis are presented.

Our study makes three key contributions. First, we introduce a novel recipient perspective for analyzing social support overload, identifying three distinct dimensions within fitness apps: emotional support overload, network support overload, and informational support overload. Second, by integrating Social Support Theory and Basic Psychological Needs Theory, our study comprehensively sheds light on the underlying mechanisms of the effect of social support overload. Social Support Theory helps us understand the effect of social support overload, while Basic Psychological Needs Theory reveals the mediating role of psychological need frustration, which could serve as the underlying mechanism of the effect of social support overload. Third, our research goes beyond the traditional scope by examining both the negative psychological and behavioral outcomes of social support overload. This dual-focus approach provides a more holistic understanding of the potential downsides of socialized fitness apps and underscores the intricate nature of IT use. In contrast to previous research that often concentrates on single facets of negative outcomes in information systems, we study both negative psychological and behavioral outcomes associated with social support overload in the fitness context to capture the complete negative effects of the dark side of fitness apps.

## 2. Theoretical Background and Hypothesis Development

### 2.1. Social Support Theory

Social support, an overarching concept in the domain of health and well-being, is defined as the perception or experience that one is cared for, esteemed, and part of a mutually supportive social network [21]. It is widely agreed that social support is a multi-dimensional construct [22]. The typology of social support proposed by Cutrona and Russel [23], including esteem, emotional, network, tangible, and informational support, has been extensively adopted and applied in the previous literature. This framework has been employed to investigate the role of different support types in promoting physical activity adherence, as well as enhancing well-being and physical health (e.g., [24,25,26]). Such widespread applications establish this typology as a valuable basis for understanding the impact of social support overload in the context of fitness apps.

While social support is widely recognized for its stress-buffering effects, which could reduce the negative effects of stresses [27], it is important to recognize that over-reception of support can actually become a source of stress and result in overload for the recipient [15,16]. We term the over-reception of social support as social support overload, which is a recipients’ perception of receiving an excessive amount of support that they are unable to effectively manage and utilize. This overload will arise when there is a mismatch between the demand and supply of social support, such as a disproportionate amount of support that cannot be reciprocated or support received that does not align with the recipient’s needs or preferences [7,23,28]. The expectation to reciprocate unwanted support would intensify the overload experienced by recipients, leading to feelings of social distress. Further, receiving excessive support may inadvertently signal a lack of competence or an inability to handle problems independently, potentially undermining the recipient’s sense of self-efficacy and respect from others [6,29]. In the context of fitness apps, where social support overload is increasingly observed, it is imperative to explore its effects and implications.

To more deeply understand social support overload in fitness apps, we tailor Cutrona and Russel’s social support typology, initially encompassing esteem, emotional, network, tangible, and informational support, to suit our study’s needs. In this typology, esteem support, which focuses on enhancing the recipient’s self-assurance and self-perception, can be seen as one manifestation of emotional support, characterized by caring, concern, encouragement, and sympathy [23,30,31]. Given their interchangeability [32], we consolidate these under the term “emotional support” to simplify and avoid potential confusion. In addition, considering the virtual nature of fitness apps, where tangible support, such as material or monetary aid, is rarely encountered, we exclude this category from our adapted typology. Consequently, we focus on three dimensions of social support overload: emotional support overload, network support overload, and informational support overload.

### 2.2. Basic Psychological Needs Theory

Basic Psychological Needs Theory, one of the six mini-theories of Self-Determination Theory (SDT), posits that social contexts, whatever their level, influence individuals by facilitating or impairing the satisfaction of basic psychological needs (BPNs) [33,34]. The theory emphasizes that individuals require three specific BPNs for optimal functioning: autonomy, which is the desire for self-determination and the freedom to make choices aligned with one’s values and interests; relatedness, the need for a sense of connection and belonging with others; and competence, the need to feel confident and capable in one’s endeavors and activities [35,36]. Satisfaction of these needs predicts positive outcomes, such as enhanced self-motivation and well-being or eudaimonia [37], while their frustration can lead to negative outcomes, including reduced subjective vitality, emotional exhaustion, and other negative affects [38]. Prior research has demonstrated that the link between social/contextual conditions and outcomes is often mediated by the degree of need satisfaction or frustration (e.g., [39,40]). Considering the role of fitness apps as a social context that can induce social support overload, our study explores how BPN frustration mediates the relationship between social support overload and outcomes such as life burnout and discontinuance.

According to the assumptions of SDT, the three BPNs of autonomy, relatedness, and competence are considered relatively independent, as they have distinct criteria for fulfillment and separate impacts on individuals’ behaviors [41,42]. Although many studies have examined overall need satisfaction without differentiating the potential effects of the three BPNs separately [43], this unidimensional view may result in a loss of valuable information, particularly regarding the theoretically expected unique explanatory power of each need, as noted by Dysvik et al. [41]. Hence, our study would investigate the BPNs’ frustration separately, focusing on autonomy need frustration, relatedness need frustration, and competence need frustration, to uncover the nuanced and individual contributions of each need.

### 2.3. Social Support Overload and BPN Frustration

#### 2.3.1. Emotional Support Overload and BPN Frustration

Emotional support overload occurs when individuals receive more emotional support than they desire [44]. In the context of fitness apps, forms of emotional support such as comments or “likes” serve to impart feelings of value, affection, and concern. However, when these supportive gestures accumulate beyond a certain threshold, they transition from being beneficial to burdensome for the recipients.

Emotional support overload can compel users to tactfully manage the decline of unwarranted support or feel obligated to reciprocate with an equitable level of support. Such dynamics can engender feelings of being controlled and a reduction in personal freedom [7,45,46]. Furthermore, internal conflicts may arise when individuals’ self-evaluation deviates from the expectations underlying the received emotional support [47]. This situation may pressure individuals to suppress their authentic thoughts and modify their behaviors to align with these perceived expectations. This strain on self-expression and self-determination would drive the recipient’s autonomy need frustration. We thus expect the following:

**H1a.** 
*Emotional support overload is positively associated with autonomy need frustration.*


Additionally, recipients inundated with emotional support might view it as insincere or superficial, lacking genuine emotional depth. Such a perception can prompt recipients to reject social support, adversely impacting the formation of meaningful connections with other users within fitness app communities. This rejection can evoke negative emotions in the support providers, such as feelings of rejection, disrespect, and lack of appreciation, subsequently leading to diminished intimacy in their interactions with the support recipients [15]. Then, the relationships between social support providers and recipients would be damaged and recipients’ relatedness needs would be frustrated. We thus expect the following:

**H1b.** 
*Emotional support overload is positively associated with relatedness need frustration.*


Moreover, an overabundance of emotional support may inadvertently impede users’ progress toward their fitness objectives, leading to a diminution in their sense of competence. As emotional support could be important for those who feel incapable of altering their circumstances [48], users intend to seek emotional support when they feel stuck in their fitness journey. However, an over-reception of emotional support may hinder their motivation to continue striving toward their fitness goals. Rather than acting as a catalyst for empowerment, excessive emotional support might instead function as a persistent reminder of the recipients’ perceived shortcomings or failures [44]. We thus expect the following:

**H1c.** 
*Emotional support overload is positively associated with competence need frustration.*


#### 2.3.2. Network Support Overload and BPN Frustration

Fitness apps offer a digital space where individuals can engage with others who have similar fitness objectives and interests, fostering a sense of community and belonging. While online social networks can enhance social connectedness, the exponential growth of personal networks can make it difficult to sustain these relationships [49,50]. This creates an imbalance between an individual’s communication needs and social abilities, ultimately resulting in network support overload. Then, the demands of social interaction, including reception, maintenance, and updates, become excessively burdensome for the recipient to manage [51].

Network support overload can result in a feeling of being constantly tethered to “always-on” communication technologies within fitness apps [52]. The constant need to check the app for fear of missing out on notifications or ongoing conversations with fellow users can generate a sense of pressure and obligation [53]. Such a state poses a threat to users’ autonomy sense, as it may restrict their freedom to disconnect and take breaks from the apps, resulting in autonomy need frustration. We thus expect the following:

**H2a.** 
*Network support overload is positively associated with autonomy need frustration.*


In addition, the reception of an overwhelming amount of network support can evoke feelings of intrusion and discomfort, as users may perceive their relationship boundaries as being violated [54]. This sense of intrusion and discomfort would prompt users to create social distance and attempt to withdraw from social relations that bring too much network support [18,55]. As this involves escaping from social relations rather than engaging in them, users’ relatedness needs are then frustrated. We thus expect the following:

**H2b.** 
*Network support overload is positively associated with relatedness need frustration.*


Furthermore, frequent exposure to workout invitations, challenges, and role models through network support in fitness apps can act as a persuasive technology feature, setting high expectations and standards for recipients [56]. This can impose a considerable demand on users, pressuring them to excel in fitness activities, which might undermine users’ confidence in their abilities and, consequently, frustrate their competence needs. Meanwhile, competence need frustration could be especially amplified when users upwardly compare themselves with more athletic individuals within their network [57]. This may bring about lower self-evaluations of appearance and body image [58,59], causing users to doubt their progress in achieving fitness goals. As a result, their competence needs may experience further frustration. We thus expect the following:

**H2c.** 
*Network support overload is positively associated with competence need frustration.*


#### 2.3.3. Informational Support Overload and BPN Frustration

Informational support overload occurs when the volume of information surpasses the recipients’ capacity to effectively process and utilize it [60]. In the context of fitness apps, this overload can lead to a state where the abundance of information, rather than being helpful, becomes a source of confusion or stress, impeding the users’ ability to benefit from the support intended to aid them in their fitness journey.

While informational support in fitness apps, such as workout guidance and nutrition tips, is generally beneficial, an overload of this support, including excessive directives and unsolicited advice, can be perceived as controlling by recipients. It may restrict users’ freedom to independently explore different options and make decisions based on personal judgment [61]. Such an overload can limit users’ autonomy in deciding on their exercise plans and controlling their fitness journey [17]. Thus, recipients’ autonomy can be impinged. We thus expect the following:

**H3a.** 
*Informational support overload is positively associated with autonomy need frustration.*


In addition, informational support overload can exhaust recipients, since they may lack the cognitive capacity to process the deluge of information [62]. Overwhelmed users might reject informational support from their social connections, leading to relatedness need frustration. Additionally, the effort to sift through excessive information can result in social network fatigue, impairing users’ ability to engage in social interactions within fitness apps [63,64]. This fatigue can further aggravate their sense of relatedness need frustration. We thus expect the following:

**H3b.** 
*Informational support overload is positively associated with relatedness need frustration.*


When informational support, which typically includes problem-solving recommendations, becomes excessive, it might inadvertently convey to recipients that they are perceived as weak or incapable of resolving issues independently [6,15]. This can lead to the frustration of users’ competence needs as they begin to doubt their ability to achieve fitness goals. In addition, informational support overload can lead to cognitive overload, hindering the effective application of this information to actionable fitness strategies and resulting in suboptimal training strategies, improper technique execution, and reduced fitness performance [65]. Consequently, feelings of failure or doubts about exercise efficacy can further drive competence need frustration. We thus expect the following:

**H3c.** 
*Informational support overload is positively associated with competence need frustration.*


### 2.4. BPN Frustration, Life Burnout, and Discontinuance

#### 2.4.1. Autonomy Need Frustration, Life Burnout, and Discontinuance

When users’ autonomy needs are frustrated, they may feel compelled by external demands, creating a tension between fulfilling these expectations and their own internal needs, without the liberty to choose freely [66]. The constant and increasing pressure can drain users’ psychological energy and deplete their self-regulatory resources in making and following fitness-independent decisions [67]. As a result, users may experience a heightened sense of mental exhaustion, which is the core element of life burnout. We thus expect the following:

**H4a.** 
*Autonomy need frustration is positively associated with life burnout.*


Moreover, autonomy need frustration can lead to the erosion of autonomous and intrinsic motivation, significantly heightening the probability of discontinuance [68,69]. The sensation of being constrained and the lack of autonomy can lead to a diminished sense of ownership over one’s fitness journey. When users begin to feel detached from a fitness app, their commitment to personal fitness goals and activities often diminishes. This decline in engagement and investment is a key factor leading to app discontinuance. We thus expect the following:

**H4b.** 
*Autonomy need frustration is positively associated with discontinuance.*


#### 2.4.2. Relatedness Need Frustration, Life Burnout, and Discontinuance

When users’ relatedness needs are frustrated, they may feel isolated, disconnected, or unsupported in their fitness journey [66]. This frustration may erode trust and create tension in relationships with other app users. Managing these strained relationships demands a higher degree of self-regulation [70], where users must put in extra effort to regulate their emotions and maintain a facade of agreeable behavior. This requirement for emotional labor can be mentally taxing [71] and can contribute to a sense of life burnout. We thus expect the following:

**H5a.** 
*Relatedness need frustration is positively associated with life burnout.*


Relatedness need frustration can also lead to discontinuance. When experiencing relatedness need frustration, users may feel disconnected and isolated. This perceived lack of belonging consequently diminishes their affective commitment to the continued use of the application [72]. In addition, relatedness need frustration can result in an unfulfilling and poor experience. Poor user experience further dampens users’ motivation and impedes sustained involvement [73]. We thus expect the following:

**H5b.** 
*Relatedness need frustration is positively associated with discontinuance.*


#### 2.4.3. Competence Need Frustration, Life Burnout, and Discontinuance

When users’ competence needs are frustrated, they may experience a sense of failure regarding the effectiveness of their exercise efforts [66]. The sense of failure could give rise to negative self-doubt and self-criticism. Consequently, users may experience a heightened psychological burden, which ultimately leads to burnout [74]. Moreover, competence need frustration is often linked with the experience of heightened stress due to performance anxiety. The constant pursuit of competence expectations may lead to vitality depletion and an elevated risk of burnout [75]. We thus expect the following:

**H6a.** 
*Competence need frustration is positively associated with life burnout.*


Furthermore, competence need frustration can cause individuals to question the value and benefits of their continued engagement with fitness apps [76]. This perception of diminished utility represents a fundamental reassessment that can significantly contribute to the decision to discontinue use. Additionally, the frustration resulting from perceived incompetence can reduce individuals’ interest in the task at hand and diminish intrinsic motivation, ultimately leading to discontinuance [77]. We thus expect the following:

**H6b.** 
*Competence need frustration is positively associated with discontinuance.*


Figure 1 illustrates the research model.

## 3. Method

### 3.1. Data Collection

Given that China accounts for the largest share of global revenue in the fitness app market [3], we decided to collect data from Chinese respondents. The inclusion criteria comprised the following: (a) current users of fitness apps; (b) those who had the ability to read and write; (c) those who consented to participate in the study. There were also exclusion criteria, including the following: (a) those who were not registered users of fitness apps; (b) those who failed our screening process; (c) those who were unwilling to consent to this study. The subjects of this study were voluntarily obtained from Credamo, a professional survey distribution platform that has an online sample pool of over 3 million respondents with diverse demographic backgrounds. The data collection was conducted through random sampling, with the online questionnaire being distributed to the platform’s own sample pool. The platform randomly assigned the survey to different participants, ensuring the randomness of the data.

To ensure the quality of data collection and reduce potential response biases, we implemented several recommended quality control methods [78]. First, we set screening questions such as “Which fitness apps have you recently used?”, “Have you interacted with other users in fitness apps?” and “Are you a registered member of the fitness apps?” to determine if the respondents were our target users. Second, attention-trap and reverse-coded questions were randomly embedded in the questionnaire. Third, we limited a reasonable completion time for the respondents to control their response time. Fourth, to encourage valid responses, each eligible respondent received a moderate monetary incentive of USD 1 for completing the survey questionnaire [79]. Fifth, we assured the anonymity of all participants and randomized the order of the survey items to reduce order effects. Finally, we removed cases with similar or missing values across all questions. In addition, the survey distribution platform also took measures to ensure the data collection quality, such as conducting screenings based on participants’ platform credit scores and historical acceptance rates, prohibiting repeated answering, and enabling intelligent verification to ensure the authenticity of responses.

We distributed the questionnaire in two stages. The data for independent variables (social support overload) and mediating variables (BPN frustration) were collected in period T_0_, and the data for dependent variables (life burnout and discontinuance) were collected in period T_1_. There was a 4-week time interval between periods T_0_ and T_1_. We collected 850 responses in period T_0_ and 475 in period T_1_. After eliminating invalid responses based on our quality control measures, a total of 443 valid samples were finally obtained. Table 1 summarizes the demographic information of these 443 samples. According to the report [80,81], the demographic attributes of the sample in our study are reasonably consistent with the population of fitness app users, which provides assurance of sample representativeness.

### 3.2. Instrument

To validate the above research model, we employed a longitudinal survey method and developed a measurement instrument to capture the involved constructs. To ensure validity, all measures were adopted from valid and reliable scales and further adapted to fit the specific context of fitness apps. Specifically, the items for the three social support overloads were adapted from Lin et al. [60], while the BPN frustration scales used in the study were adapted from James et al. [66]. In addition, we adapted items for measuring life burnout from Whelan and Clohessy [82] and items for measuring discontinuance from Zhang et al. [64].

All items were measured on a five-point Likert scale ranging from strongly disagree (1) to strongly agree (5). Since we collected data in China and the original items were in English, we used the back-translation method to convert these items into Chinese [83]. After initially making up the questionnaire, a pilot test was conducted among 50 postgraduate students with experience using fitness apps such as KEEP, Nike+, Strava, or any other. Based on respondent feedback and the factor analysis results, we modified items and decided on the measurement instrument. The detailed scale is presented in Table 2.

## 4. Results

We used SmartPLS version 4.0 to conduct our analysis, employing partial least squares (PLS) to analyze the measurement quality and the path model. PLS–SEM emphasizes prediction and facilitates explaining the model’s relationships by maximizing the endogenous latent variables’ explained variance [84]. It can explore the relationships between multiple independent and dependent variables [85]. With the inclusion of several reflective measured constructs in our path model, PLS–SEM is a suitable technique for our research. Following a two-step analysis procedure, we first examined the measurement model and then analyzed the structural model [86].

### 4.1. Measurement Model

#### 4.1.1. Reliability and Validity

The measurement model was examined to assess the variables’ reliability, convergent validity, and discriminant validity. Table 3 illustrates the factor loadings, Cronbach’s alpha (Cronbach’s α), composite reliability (CR), and average variance extracted (AVE) of all of the constructs in the model. As shown in Table 3, the Cronbach’s α and CR values of all of the variables meet the recommended threshold values of 0.7 [87]. Thus, the reliability of the measurement model is desirable. Meanwhile, the item loadings for all construct items are above 0.7, and all AVE values are above 0.5, thus confirming the good convergent validity [86]. Furthermore, the diagonal values in Table 4, representing the square roots of AVEs for each construct, indicate good discriminant validity, as they all exceed the correlations [88]. Similarly, the heterotrait–monotrait ratio (HTMT) in Table 5, with all values being below 0.9, provides additional evidence of the satisfactory discriminant validity [89].

#### 4.1.2. Common Method Bias

Common method bias (CMB) may occur in studies that collect data from the same source or rater [90]. To assess CMB, we conducted two tests. The correlation matrix showed no high correlation between the factors, with the highest correlation being r = 0.733, which was below the threshold for extremely high correlation (r > 0.90) [91], indicating a low possibility of CMB. Additionally, we used a marker variable, “blue attitude”, which was theoretically unrelated to the main study variables [92]. The analysis found no significant association between “blue attitude” and our two dependent variables, with correlations ranging from 0.032 to −0.156 and averaging 0.0997. Comparing the baseline model without the marker variable to the CMB test model revealed no significant differences in path coefficients and explained variance, as shown in Table 6. These statistical analysis results suggest that CMB is not a serious concern in our study.

#### 4.1.3. Social Desirability Bias

Social desirability bias (SDB) is a form of response bias. In survey research, there is a risk of respondents providing socially desirable responses, potentially distorting the true relationships between variables [90,93]. To assess the presence of SDB, we administered a 13-item scale [94] and performed a correlation analysis between all variables and SDB scores. The results revealed that only one correlation exceeded the 0.40 threshold (r = −0.444), suggesting a modest influence of SDB on the participants’ responses.

#### 4.1.4. Multicollinearity

To assess the potential presence of multicollinearity, variance inflation factors (VIFs) were examined. Table 7 presents the inner and outer VIFs for the items included in our model. It was observed that all VIF values are below 5.0, indicating the absence of significant multicollinearity concerns in our model [95].

### 4.2. Structural Model

We analyzed the structural model by using the bootstrapping procedure. Table 8 and Figure 2 summarize the results of our structural model. The calculated SRMR is 0.044, exhibiting a high level of model fit of our research model [96].

The analysis reveals that emotional support overload does not demonstrate a significant relationship with autonomy need frustration and, instead, shows a negative association with relatedness need frustration and competence need frustration. As such, the analysis results reject H1a, H1b, and H1c. Conversely, the findings support our hypotheses that network support overload and informational support overload have positive impacts on BPN frustration, providing full support for H2 and H3. Regarding autonomy need frustration and relatedness need frustration, the study found that they are significantly associated with both life burnout and discontinuance, giving complete support for H4 and H5. Additionally, while competence need frustration significantly influences life burnout, it does not affect discontinuance, supporting H6a but not H6b. Notably, regarding the explanatory power of the research model, the R^2^ results are all acceptable and are as follows: autonomy need frustration (R^2^ = 0.368), relatedness need frustration (R^2^ = 0.108), competence need frustration (R^2^ = 0.189), life burnout (R^2^ = 0.421), and discontinuance (R^2^ = 0.361).

### 4.3. Post Hoc Analysis

#### 4.3.1. Mediation Analysis

To examine whether BPN frustration mediates the relationship between social support overload and life burnout/discontinuance, our study used the bootstrapping method to test for mediation using SmartPLS [97]. The mediation analysis results are presented in Table 9. The results highlight the significant mediating role of BPN frustration in the relationship between social support overload and life burnout/discontinuance. Specifically, autonomy need frustration fully mediates the impact of informational support overload on life burnout and discontinuance, and it partially mediates the effects of network support overload. Similarly, relatedness need frustration fully mediates the influence of emotional and informational support overload on life burnout and discontinuance, and it partially mediates the effects of network support overload. Furthermore, competence need frustration fully mediates the connection between emotional and informational support overload and burnout, and it partially mediates the effects of network support overload.

#### 4.3.2. Multigroup Analysis

We then conducted a multigroup analysis (MGA) on the control variables with significant results (the gender and the proficiency groups) to seek better insight into the social support of fitness apps. Regarding proficiency, a dummy variable was used to classify users as proficient (users who rated their level of fitness app use as advanced or expert) and nonproficient (novice or intermediate). The MGA starts with the Measurement Invariance of Composite Models (MICOM) procedure to assess invariance. Any finding of a between-group difference is questionable if measurement invariance is not demonstrated [85]. The results of the MICOM procedure are presented in Appendix A, which shows partial measurement invariance, allowing for MGA to be performed. In this study, MGA was performed by comparing the corresponding paths of different groups.

Table 10 reports the significant paths in the MGA. Specifically, our findings indicate that women may be more affected by informational support overload, autonomy need frustration, and relatedness need frustration than men. In addition, proficient users may be more affected by emotional support overload, autonomy need frustration, and relatedness need frustration than non-proficient users.

## 5. Discussion

To explore how social support overload impacts users’ life burnout and discontinuance in fitness apps, we establish and validate a research model based on Social Support Theory and Basic Psychological Needs Theory. Generally, we find that recipients’ social support overload affects life burnout and discontinuance through the frustration of their basic psychological needs. Some of the hypotheses are not supported, and we interpret them as follows.

Regarding the insignificant relationship between emotional support overload and autonomy need frustration, it suggests that excessive emotional support does not contribute to feelings of autonomy frustration. The possible reason for this result could be attributed to the characteristics of the main user group of fitness apps. The main users of fitness apps are young and middle-aged users who possess strong self-regulation and coping skills, enabling them to effectively handle emotional support overload [98] and then mitigate the autonomy need frustration. Furthermore, it is possible that the positive effects of receiving ample emotional support, such as a sense of validation and encouragement [99], may outweigh the negative impacts of emotional support overload. As a result, the overall emotional support may help users feel more empowered, counteracting any potential frustration related to autonomy.

In addition, contrary to the initial hypothesis, emotional support overload exhibited negative effects on both relatedness and competence need frustration. These results imply that a high quantity of emotional support may have a strong soothing effect in alleviating frustrations [100]. For relatedness need frustration, an abundance of emotional support can also serve as an indicator of a robust support network, which helps decrease the probability of experiencing relatedness need frustration. This suggests that, in fitness apps, where users often engage with others and receive support, the feeling of being connected to a larger social network can reduce feelings of isolation and help fulfill the need for relatedness, even when emotional support is perceived as excessive. Regarding competence need frustration, emotional support pertains to nurturant support, which is typically provided when users in fitness apps face stress or encounter difficulties in achieving competence. Therefore, excessive emotional support may relieve the stress and then alleviate competence need frustration to some degree [15]. Essentially, the emotional support provided could help users manage the pressures that they face while striving for personal competence, thus reducing feelings of inadequacy or failure in their fitness journey.

Furthermore, the study found an insignificant effect of competence need frustration on discontinuance, indicating that users may continue using fitness apps despite experiencing competence need frustration. This can be explained as follows; compared with the frustration of the other two BPNs, users of fitness apps can recover their sense of competence by exerting more effort in the exercise tasks, which, in turn, motivates them to persist in using the fitness apps [101,102]. However, simply putting more effort into exercise tasks may not be able to restore users’ relatedness and autonomy.

### 5.1. Theoretical Implications

Our study provides three theoretical implications. First, a novel recipient perspective was applied to explore the dimensions of social support overload in fitness apps. While the previous literature has primarily focused on “social overload” from a unidimensional and provider perspective [103], we take a multidimensional and recipient perspective to understand social support overload more precisely and completely. To contextualize social support overload [104], three dimensions of social support overload in fitness apps are identified: emotional support overload, network support overload, and informational support overload. The role of these three social support overload dimensions is demonstrated in this study.

Moreover, by integrating Social Support Theory and Basic Psychological Needs Theory, our study comprehensively sheds light on the working mechanism of the effect of social support overload. By incorporating Social Support Theory, we not only classify social support overload but also gain insights into its role. In addition, Basic Psychological Needs Theory helps us uncover the mediating role of psychological need frustration, which could serve as the underlying mechanism of the effect of social support overload. Since both theories emphasize the role of social contexts, the integration of Social Support Theory and Basic Psychological Needs Theory could mutually complement each other, enabling a complete understanding of the working mechanisms behind the effects of social support overload [73].

Lastly, our study enhances the understanding of the potential downsides of socialized fitness apps by examining both negative psychological and behavioral outcomes. This approach marks a departure from previous research that primarily focused on a single negative outcome related to information systems [10,60,64]. By investigating the effects of social support overload in fitness apps on both life burnout and discontinuance, we provide a comprehensive view of the negative impacts, contributing to a deeper understanding of the complex nature of IT usage and its implications for user well-being and engagement. This comprehensive analysis aims to offer nuanced insights into the effects and intricacies of socialized fitness app usage.

### 5.2. Practical Implications

The findings of our study also have some practical implications. First, to mitigate social support overload among fitness app users, actions can be taken by both users and app managers. Users can be guided to adopt self-management strategies, such as regulating interaction frequency and filtering notifications, to manage their social support intake. In addition, fitness app managers can foster a supportive and balanced app environment by providing customization options that align with individual preferences and using intelligent algorithms to ensure that the information presented is pertinent and non-overwhelming. These measures can help prevent the potential overload of network support and informational support, which are identified as significant contributors to users’ BPN frustration.

In addition, given the mediating role of BPN frustration, interventions can be developed to target its reduction. For autonomy need frustration, fitness app managers can provide users with adjustable settings, such as those for workout duration, intensity levels, and rest intervals, to grant users greater control over their fitness routines. Advanced technologies such as AI-driven algorithms and backend data analysis can be employed to make personal recommendations based on users’ activity data. For the frustration of relatedness and competence needs, incorporating gamification elements would be effective. Implementing leaderboards or team-based challenges in fitness apps can foster friendly competition and a sense of belonging among users. Meanwhile, participation in leaderboards and challenges can be made optional, giving users the freedom to engage with these features at their discretion. Virtual badges or points can be used as rewards to reinforce users’ sense of competence when they achieve milestones in their fitness journey. Furthermore, offering challenging yet achievable exercise activities that gradually increase in difficulty allows users to challenge themselves while still feeling accomplished.

Lastly, recognizing the impact of gender and proficiency of use, interventions can be tailored to mitigate discontinuance rates in fitness app usage. Building upon the stronger positive association between relatedness need frustration and discontinuance among female and nonproficient users, fitness app managers can make use of their database of registered users to recommend compatible peers with similar backgrounds to women and nonproficient users. Fitness app managers can implement measures to guide the nature of interactions. This includes establishing clear group rules that promote supportive, accurate, and constructive communication, as well as fostering a positive community culture. Additionally, a feedback mechanism could allow users to evaluate the quality of their social interactions, enabling managers to refine grouping algorithms and better tailor peer recommendations. By facilitating the establishment of social relationships among these user groups, fitness app managers can effectively address users’ relatedness need frustration and enhance their engagement with fitness apps.

### 5.3. Limitations and Future Research Directions

Several limitations to our study suggest potential avenues for future research. First, the generalizability of our study results could be improved. Our research was conducted with a specific cultural and demographic sample, which may limit the applicability of the findings to broader populations. Further research could examine the cross-cultural differences in the effects of social support overload among other user groups with different characteristics.

Moreover, there are additional outcomes that can be explored to deepen our understanding of the role of social support overload in fitness apps. Except for life burnout and discontinuance, future research could consider investigating other behavioral and psychological outcomes. Meanwhile, incorporating physiological outcomes, such as changes in weight, body fat percentage, duration, and frequency of actual exercise behaviors, could provide a more direct and objective assessment of individuals’ fitness app outcomes.

Additionally, while total discontinuance serves as the ultimate behavioral outcome, the process leading to discontinuance may vary among users. It would be valuable to explore whether users experience a gradual burnout or whether the overload of social support leads to immediate abandonment of the app. This distinction could provide deeper insights into user behavior over time. Future research could benefit from adopting longitudinal or time-lagged study designs to track changes in app use over extended periods, capturing the dynamic nature of user engagement and the gradual impact of factors such as social support overload.

Finally, it is important to explore additional mechanisms underlying the effects of social support overload on the negative psychological and behavioral outcomes of fitness app users. Although our mediating analysis reveals the role of BPN frustration, it is worth noting that some focal relationships are partially mediated by BPNs. Future studies can also investigate how factors such as social support, autonomy, and self-efficacy might interact with these negative factors, offering further insights into how these factors influence the experience of overload. Therefore, other potential mechanisms can be considered based on valid theoretical perspectives in future studies. Additionally, deeper investigations are warranted to develop and evaluate app design strategies that effectively tackle social support overload in fitness apps.

## 6. Conclusions

By integrating Social Support Theory and Basic Psychological Needs Theory, our study presents a comprehensive theoretical framework for examining the relationship between social support overload and users’ life burnout and discontinuance in the context of fitness apps. Our findings highlight the mediating role of BPN frustration in the above relationships. Specifically, network support overload and informational support overload demonstrate positive associations with BPN frustration, while emotional support overload yields mixed results. The frustration of all three BPNs significantly contributes to life burnout, while the frustration of only autonomy and relatedness needs significantly impacts discontinuance. Through our novel recipient perspective, our study provides a nuanced and deep understanding of the impact of social support overload in fitness apps. Additionally, many effective practical recommendations for fitness app users and managers can be derived.

## Figures and Tables

**Figure 1 healthcare-13-00191-f001:**
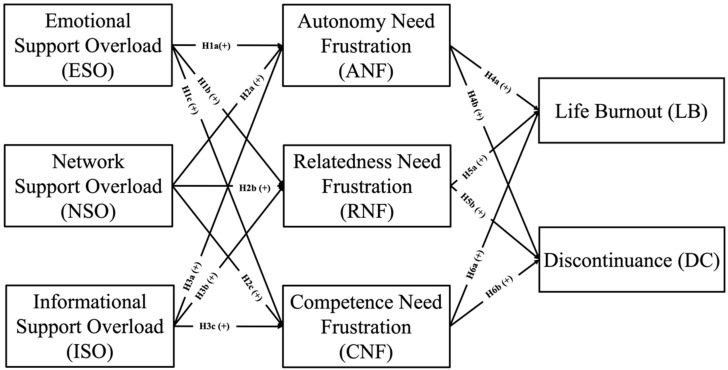
The research model.

**Figure 2 healthcare-13-00191-f002:**
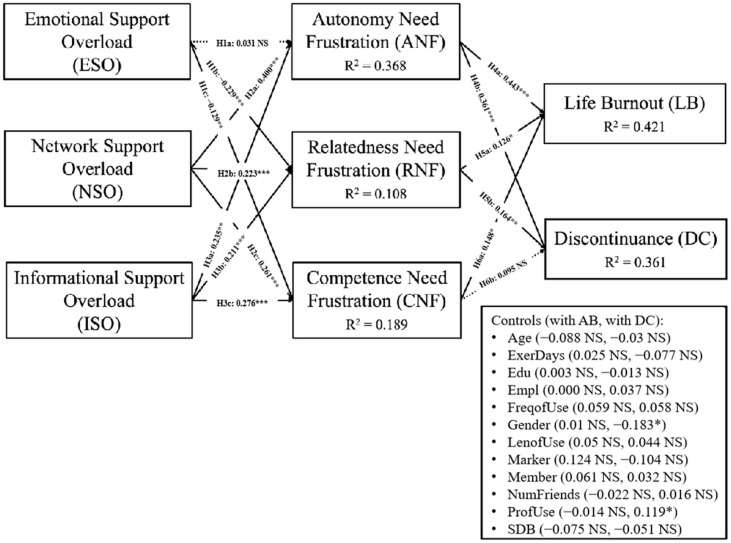
Structural model results. Note: *** *p* < 0.001, ** *p* < 0.01, * *p* < 0.05. NS represents not significant.

**Table 1 healthcare-13-00191-t001:** Demographic information of the sample (*n* = 443).

Characteristics	Category	Statistics, *n* (%)
Age	18–30 Years	262 (59.1)
	31–40 Years	146 (33.0)
	41–50 Years	22 (5.0)
	51–60 Years	12 (2.7)
	61 Years or More	1 (0.2)
Gender	Male	172 (38.8)
	Female	271 (61.2)
Membership	Member	323 (72.9)
	Non-Member	120 (27.1)
Level of Education	Junior High School and Below	2 (0.5)
	High School/Vocational–Technical Education	13 (2.9)
	Junior College	41 (9.3)
	Bachelor’s Degree	337 (76.0)
	Master’s Degree and Above	50 (11.3)
Employment Status	Employed Full-Time	349 (78.8)
	Employed Part-Time	3 (0.7)
	Self-Employed	14 (3.1)
	Student	75 (16.9)
	Not Employed	2 (0.5)
Exercise Days per Week	0~3 Days	129 (29.1)
	4~7 Days	314 (70.9)
Length of Fitness App Use	Less than 6 Months	10 (2.2)
	6 Months to 1 Year (Including 6 Months)	73 (16.5)
	1–2 Years (Including 1 Year)	167 (37.7)
	2–4 Years (Including 2 Years)	136 (30.7)
	4 Years or More	57 (12.9)
Frequency of Fitness App Use	Multiple Times per Day	62 (14.0)
	Once per Day	103 (23.2)
	Multiple Times per Week	269 (60.7)
	Once per Week	6 (1.4)
	Multiples Times per Month	3 (0.7)
	Once per Month	0 (0.0)
	Less than Once per Month	0 (0.0)
Proficiency in Fitness App Use	Novice	21 (4.8)
	Intermediate	217 (49.0)
	Advanced	180 (40.6)
	Expert	25 (5.6)
Number of Exercise Friends	Less than 20	113 (25.5)
	20–40	125 (28.2)
	41–60	86 (19.4)
	61–80	44 (9.9)
	81–100	39 (8.8)
	More than 100	36 (8.2)

**Table 2 healthcare-13-00191-t002:** Measurement item details.

Constructs	Items	Questionnaire Items
Emotional Support Overload	ESO1	My friends on the fitness apps care for me too often.
	ESO2	My friends on the fitness apps care about my feelings and emotions too much.
	ESO3	My friends on the fitness apps take too much care of my well-being.
	ESO4	My friends on the fitness apps pay too much attention to my training.
Network Support Overload	NSO1	I receive more communication messages and social requests from friends on the fitness apps than I can process.
	NSO2	I feel compelled to engage more on fitness apps (posting, liking, commenting) to maintain social networks.
	NSO3	I feel overloaded with communication and connection.
	NSO4	Dealing with my friends’ problems on fitness apps is too burdensome.
Informational Support Overload	ISO1	I am often distracted by excessive suggestions and advice about exercise from the fitness apps.
	ISO2	I am overwhelmed by the number of suggestions and advice about exercise that I process daily from the fitness apps.
	ISO3	I feel some problems with too many suggestions and advice about exercise from the fitness apps to synthesize.
Autonomy Need Frustration	ANF1	Using fitness apps makes me feel forced to exercise in ways I wouldn’t normally choose.
	ANF2	Using fitness apps makes me feel pressured to do exercise.
Relatedness Need Frustration	RNF1	I feel excluded from friends on the fitness apps.
	RNF2	I feel that friends on the fitness apps are cold and distant toward me.
	RNF3	I feel like friends on the fitness apps dislike me.
	RNF4	I feel like the relationships I have with friends on the fitness apps are superficial.
Competence Need Frustration	CNF1	After using the fitness apps, I often doubt my ability to exercise well.
	CNF2	After using the fitness apps, I feel disappointed with my exercise performance.
	CNF3	After using the fitness apps, I feel insecure about my exercise abilities.
	CNF4	After using the fitness apps, I feel like a failure because of the exercise mistakes I make.
Life Burnout	LB1	After using the fitness apps, I feel tired.
	LB2	After using the fitness apps, I feel physically exhausted.
	LB3	After using the fitness apps, I think “I cannot take it anymore”.
	LB4	After using the fitness apps, I feel worn out.
Discontinuance	DC1	I use my current fitness apps far less than today.
	DC2	I sometimes take a short break from the fitness apps and return later.
	DC3	I discontinue the use of the fitness apps.

**Table 3 healthcare-13-00191-t003:** Measurement quality model.

Construct	Items	Loadings	Cronbach’s α	CR	AVE
Autonomy Need Frustration (ANF)	ANF1	0.904	0.793	0.906	0.828
	ANF2	0.916			
Competence Need Frustration (CNF)	CNF1	0.897	0.881	0.918	0.737
	CNF2	0.848			
	CNF3	0.882			
	CNF4	0.805			
Discontinuance (DC)	DC1	0.856	0.775	0.870	0.690
	DC2	0.781			
	DC3	0.854			
Life Burnout (LB)	LB1	0.870	0.864	0.908	0.711
	LB2	0.836			
	LB3	0.814			
	LB4	0.853			
Emotional Support Overload (ESO)	ESO1	0.856	0.892	0.925	0.755
	ESO2	0.878			
	ESO3	0.871			
	ESO4	0.871			
Informational Support Overload (ISO)	ISO1	0.907	0.900	0.938	0.834
	ISO2	0.905			
	ISO3	0.927			
Network Support Overload (NSO)	NSO1	0.857	0.901	0.931	0.772
	NSO2	0.864			
	NSO3	0.903			
	NSO4	0.889			
Relatedness Need Frustration (RNF)	RNF1	0.839	0.800	0.869	0.625
	RNF2	0.797			
	RNF3	0.762			
	RNF4	0.762			

**Table 4 healthcare-13-00191-t004:** Correlation analysis of latent variables and the square root of the AVE.

	ANF	CNF	DC	LB	Marker	ESO	ISO	NSO	RNF
ANF	**0.91**								
CNF	0.546	**0.859**							
DC	0.501	0.443	**0.831**						
LB	0.583	0.499	0.733	**0.843**					
Marker	−0.137	−0.142	−0.155	−0.074	**0.868**				
ESO	0.376	0.149	0.218	0.223	0.033	**0.869**			
ISO	0.525	0.401	0.369	0.418	−0.102	0.433	**0.913**		
NSO	0.581	0.374	0.442	0.475	−0.057	0.607	0.691	**0.878**	
RNF	0.299	0.559	0.383	0.382	−0.147	−0.002	0.266	0.23	**0.791**

Notes: The bold numbers represent the square root of the AVE.

**Table 5 healthcare-13-00191-t005:** Heterotrait–monotrait ratio (HTMT).

	ANF	CNF	DC	LB	Marker	ESO	ISO	NSO
ANF								
CNF	0.653							
DC	0.638	0.535						
LB	0.704	0.571	0.898					
Marker	0.162	0.154	0.186	0.087				
ESO	0.444	0.168	0.263	0.251	0.059			
ISO	0.62	0.45	0.443	0.471	0.114	0.481		
NSO	0.684	0.417	0.526	0.535	0.068	0.677	0.768	
RNF	0.371	0.665	0.483	0.46	0.169	0.061	0.306	0.262

**Table 6 healthcare-13-00191-t006:** PLS marker variable approach.

Path Coefficients	Baseline Model Without Marker Variable	CMB Test Model with Marker Variable
Emotional Support Overload → Autonomy Need Frustration	0.031 (NS)	0.031 (NS)
Emotional Support Overload → Relatedness Need Frustration	−0.229 ***	−0.229 ***
Emotional Support Overload → Competence Need Frustration	−0.129 **	−0.129 **
Network Support Overload → Autonomy Need Frustration	0.400 ***	0.400 ***
Network Support Overload → Relatedness Need Frustration	0.223 ***	0.223 ***
Network Support Overload → Competence Need Frustration	0.261 ***	0.261 ***
Informational Support Overload → Autonomy Need Frustration	0.235 **	0.235 **
Informational Support Overload → Relatedness Need Frustration	0.211 ***	0.211 ***
Informational Support Overload → Competence Need Frustration	0.276 ***	0.276 ***
Autonomy Need Frustration → Life Burnout	0.441 ***	0.443 ***
Autonomy Need Frustration → Discontinuance	0.363 ***	0.361 ***
Relatedness Need Frustration → Life Burnout	0.124 *	0.126 *
Relatedness Need Frustration → Discontinuance	0.166 **	0.164 **
Competence Need Frustration → Life Burnout	0.147 *	0.148 *
Competence Need Frustration → Discontinuance	0.095 (NS)	0.095 (NS)
Marker Variable → Life Burnout	N/A	0.124 (NS)
Marker Variable → Discontinuance	N/A	−0.104 (NS)
Explanatory power (R^2^)		
Autonomy Need Frustration	36.8%	36.8%
Relatedness Need Frustration	10.8%	10.8%
Competence Need Frustration	18.9%	18.9%
Life Burnout	41.9%	42.1%
Discontinuance	35.9%	36.1%

Note: *** *p* < 0.001, ** *p* < 0.01, * *p* < 0.05. NS represents not significant.

**Table 7 healthcare-13-00191-t007:** Collinearity statistics (VIF).

Construct (Inner VIF)	Item	Outer VIF	Construct (Inner VIF)	Item	Outer VIF
ANF (1.581)	ANF1	1.757	ESO (1.585)	ESO1	2.142
	ANF2	1.757		ESO2	2.629
CNF (2.160)	CNF1	2.837		ESO3	2.582
	CNF2	2.142		ESO4	2.346
	CNF3	2.698	ISO (1.915)	ISO1	2.778
	CNF4	1.806		ISO2	2.629
DC	DC1	1.928		ISO3	3.143
	DC2	1.359	NSO (2.466)	NSO1	2.356
	DC3	1.877		NSO2	2.236
LB	LB1	2.399		NSO3	3.063
	LB2	2.099		NSO4	2.891
	LB3	1.909	RNF (1.673)	RNF1	1.787
	LB4	2.255		RNF2	1.705
Marker (1.117)	Marker1	2.436		RNF3	1.490
	Marker2	2.332		RNF4	1.590
	Marker3	1.660			

**Table 8 healthcare-13-00191-t008:** Detailed results of tested hypotheses (*n* = 434).

Tested Path	Path Coefficient (Sample Mean)	Std. Dev.	β	t-Statistic	*p*-Value	Results
H1a: Emotional Support Overload → Autonomy Need Frustration	0.033	0.043	0.031	0.723	0.47 NS	Not Supported
H1b: Emotional Support Overload → Relatedness Need Frustration	−0.23	0.049	−0.229	4.723	0.000 ***	Not Supported
H1c: Emotional Support Overload → Competence Need Frustration	−0.129	0.048	−0.129	2.697	0.007 **	Not Supported
H2a: Network Support Overload →Autonomy Need Frustration	0.4	0.07	0.4	5.746	0.000 ***	Supported
H2b: Network Support Overload → Relatedness Need Frustration	0.225	0.062	0.223	3.589	0.000 ***	Supported
H2c: Network Support Overload → Competence Need Frustration	0.262	0.063	0.261	4.141	0.000 ***	Supported
H3a: Informational Support Overload → Autonomy Need Frustration	0.234	0.069	0.235	3.402	0.001 **	Supported
H3b: Informational Support Overload → Relatedness Need Frustration	0.211	0.06	0.211	3.496	0.000 ***	Supported
H3c: Informational Support Overload → Competence Need Frustration	0.276	0.057	0.276	4.805	0.000 ***	Supported
H4a: Autonomy Need Frustration → Life Burnout	0.445	0.053	0.443	8.363	0.000 ***	Supported
H4b: Autonomy Need Frustration → Discontinuance	0.362	0.059	0.361	6.076	0.000 ***	Supported
H5a: Relatedness Need Frustration → Life Burnout	0.128	0.059	0.126	2.145	0.032 *	Supported
H5b: Relatedness Need Frustration → Discontinuance	0.162	0.059	0.164	2.771	0.006 **	Supported
H6a: Competence Need Frustration → Life Burnout	0.147	0.064	0.148	2.323	0.020 *	Supported
H6b: Competence Need Frustration → Discontinuance	0.097	0.063	0.095	1.505	0.132 NS	Not Supported
Member → Life Burnout	0.061	0.11	0.061	0.559	0.576 NS	/
Member → Discontinuance	0.032	0.113	0.032	0.286	0.775 NS	/
ExerDays → Life Burnout	0.024	0.053	0.025	0.473	0.636 NS	/
ExerDays → Discontinuance	−0.077	0.052	−0.077	1.475	0.140 NS	/
LenofUse → Life Burnout	0.05	0.054	0.05	0.918	0.359 NS	/
LenofUse → Discontinuance	0.044	0.061	0.044	0.724	0.469 NS	/
FreofUse → Life Burnout	0.057	0.044	0.059	1.333	0.183 NS	/
FreofUse → Discontinuance	0.058	0.047	0.058	1.227	0.220 NS	/
ProofUse → Life Burnout	−0.016	0.052	−0.014	0.269	0.788 NS	/
ProofUse → Discontinuance	−0.116	0.054	−0.116	2.135	0.033 *	/
NumFriends → Life Burnout	−0.023	0.053	−0.022	0.412	0.681 NS	/
NumFriends → Discontinuance	0.015	0.054	0.016	0.295	0.768 NS	/
Gender → Life Burnout	0.012	0.078	0.01	0.127	0.899 NS	/
Gender → Discontinuance	−0.184	0.083	−0.183	2.218	0.027 *	/
Age → Life Burnout	−0.088	0.052	−0.088	1.686	0.092 NS	/
Age → Discontinuance	−0.029	0.051	−0.03	0.601	0.548 NS	/
Empl → Life Burnout	0.003	0.059	0	0.001	0.999 NS	/
Empl → Discontinuance	0.055	0.056	0.055	0.988	0.323 NS	/
Edu → Life Burnout	0.002	0.042	0.003	0.06	0.952 NS	/
Edu → Discontinuance	−0.014	0.043	−0.013	0.307	0.759 NS	/
Salary → Life Burnout	0.009	0.067	0.007	0.108	0.914 NS	/
Salary → Discontinuance	0.118	0.069	0.119	1.726	0.084 NS	/
Marker → Life Burnout	0.116	0.096	0.124	1.297	0.195 NS	/
Marker → Discontinuance	−0.113	0.09	−0.104	1.16	0.246 NS	/
SDB → Life Burnout	−0.072	0.052	−0.075	1.444	0.149 NS	/
SDB → Discontinuance	−0.048	0.05	−0.051	1.011	0.312 NS	/

Note: *** *p* < 0.001, ** *p* < 0.01, * *p* < 0.05. NS represents not significant.

**Table 9 healthcare-13-00191-t009:** Results of the mediation test.

Proposed Relationship			Mediation Test (ab) (Indirect Effects)			Full/Partial Mediation Test (c′)			Mediation Effect
IV	M	DV	2.5%	97.5%	Include Zero?	2.5%	97.5%	Include Zero?	
ESO	ANF	LB	−0.018	0.042	Yes	−0.128	0.044	Yes	None
ESO	ANF	DC	−0.012	0.031	Yes	−0.037	0.134	Yes	None
ESO	RNF	LB	−0.056	0.000	No	−0.128	0.044	Yes	Full
ESO	RNF	DC	−0.063	−0.008	No	−0.037	0.134	Yes	Full
ESO	CNF	LB	−0.034	−0.002	No	−0.128	0.044	Yes	Full
ESO	CNF	DC	−0.029	0.007	Yes	−0.037	0.134	Yes	None
ISO	ANF	LB	0.033	0.132	No	−0.08	0.132	Yes	Full
ISO	ANF	DC	0.018	0.099	No	−0.112	0.129	Yes	Full
ISO	RNF	LB	0.000	0.054	No	−0.08	0.132	Yes	Full
ISO	RNF	DC	0.006	0.067	No	−0.112	0.129	Yes	Full
ISO	CNF	LB	0.000	0.076	No	−0.08	0.132	Yes	Full
ISO	CNF	DC	−0.013	0.06	Yes	−0.112	0.129	Yes	None
NSO	ANF	LB	0.073	0.212	No	0.08	0.318	No	Partial
NSO	ANF	DC	0.037	0.16	No	0.052	0.316	No	Partial
NSO	RNF	LB	0.000	0.059	No	0.08	0.318	No	Partial
NSO	RNF	DC	0.006	0.071	No	0.052	0.316	No	Partial
NSO	CNF	LB	0.000	0.073	No	0.08	0.318	No	Partial
NSO	CNF	DC	−0.014	0.057	Yes	0.052	0.316	No	None

**Table 10 healthcare-13-00191-t010:** Results of the multigroup analysis.

Group	Paths	β_0_	*p*-Value	β_1_	*p*-Value	Coefficient Difference	*p*-Value
Male vs. Female	H3a: ISO→ANF	0.033	0.735	0.365	0.000 ***	−0.332	0.014 *
	H4b: ANF→DC	0.516	0.000 ***	0.236	0.001 **	0.281	0.016 *
	H5b: RNF→DC	0.067	0.379 NS	0.3	0.000 ***	−0.233	0.022 *
Proficient vs. Nonproficient	H1c: ESO→CNF	−0.174	0.001 **	0.048	0.543 NS	−0.222	0.022 *
	H4a: ANF→LB	0.558	0.000 ***	0.283	0.000 ***	0.275	0.007 **
	H5b: RNF→DC	0.054	0.453 NS	0.26	0.001 **	−0.205	0.048 *

Note: *** *p* < 0.001, ** *p* < 0.01, * *p* < 0.05. NS represents not significant.

## Data Availability

The dataset is available from the authors upon request. The raw data supporting the conclusions of this article will be made available by the authors upon request.

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
