# Peer review of "When I Receive Too Much Social Support: The Effect of Social Support Overload on Users’ Life Burnout and Discontinuance in Fitness Apps"

_healthcare, 2025, doi:10.3390/healthcare13020191_

Round 1
Reviewer 1 Report
Comments and Suggestions for Authors
Dear Authors,
Here are the recommended changes for your paper:
Clearly explain theoretically the negative results achieved with emotional overload of support.
Also, the methods used to suppress the response biases in data collection can also be explained to improve the readability of the manuscript.
Author Response
Comment1:
Clearly explain theoretically the negative results achieved with emotional overload of support.
Response1:
Thanks for your comments! We have clarified the negative results achieved with emotional support overload in the discussion section as follows [page: 18, line: 773-798]:
“Regarding the insignificant relationship between emotional support overload and autonomy need frustration, it suggests that excessive emotional support does not contribute to feelings of autonomy frustration. The possible reason for this result could be attributed to the characteristics of the main user group of fitness apps. The main users of fitness apps are young and middle-aged users who possess strong self-regulation and coping skills, enabling them to effectively handle emotional support overload [1] and then mitigate the autonomy need frustration. Furthermore, it is possible that the positive effects of receiving ample emotional support, such as a sense of validation and encouragement [2], may outweigh the negative impacts of emotional support overload. As a result, the overall emotional support may help users feel more empowered, counteracting any potential frustration related to autonomy.
In addition, contrary to the initial hypothesis, emotional support overload exhibited negative effects on both relatedness and competence need frustration. These results imply that a high quantity of emotional support may have a strong soothing effect in alleviating frustrations [3]. For relatedness need frustration, an abundance of emotional support can also serve as an indicator of a robust support network which helps decrease the probability of experiencing relatedness need frustration. This suggests that, in fitness apps, where users often engage with others and receive support, the feeling of being connected to a larger social network can reduce feelings of isolation and help fulfill the need for relatedness, even when emotional support is perceived as excessive. Regarding competence need frustration, emotional support pertains to nurturant support which is typically provided when users in fitness apps face stress or encounter difficulties in achieving competence. Therefore, excessive emotional support may relieve the stress, and then alleviate competence need frustration to some degree [4]. Essentially, the emotional support provided could help users manage the pressures that they face while striving for personal competence, thus reducing feelings of inadequacy or failure in their fitness journey.”
Comment2:
Also, the methods used to suppress the response biases in data collection can also be explained to improve the readability of the manuscript.
Response2:
Thank you for your thoughtful feedback. We have used several methods to suppress and estimate response biases.
First, one effective method of estimating response biases is by comparing the demographic information of our sample with the general population [5]. In our study, the demographic characteristics of the participants are generally consistent with the broader population of fitness app users. We have demonstrated it in the data collection section [page: 9, line: 502-505]:
“According to the report [6,7], the demographic attributes of the sample in our study is reasonably consistent with the population of fitness apps users, which provides assurance of sample representativeness.”
Second, we also implemented several data quality control measures, such as attention-trap questions, randomization of item order, and the exclusion of invalid responses, to minimize potential response biases, such as acquiescence bias, leniency bias, etc. [8,9]. We have demonstrated it in the data collection section [page: 8, line: 471-485]:
“To ensure the quality of data collection and reduce potential response biases, we implemented several recommended quality control methods [10]. First, we set screening questions such as “Which fitness apps have you recently used?”, “Have you interacted with other users in fitness apps?” and “Are you a registered member of the fitness apps?” to determine if the respondents were our target users. Second, attention-trap and reverse-coded questions were randomly embedded in the questionnaire. Third, we limited a reasonable completion time for the respondents to control their response time. Fourth, to encourage valid responses, each eligible respondent received a moderate monetary incentive of one dollar for completing the survey questionnaire [11]. Fifth, we assured the anonymity of all participants and randomized the order of the survey items to reduce order effects. Finally, we removed cases with similar or missing values across all questions. In addition, the survey distribution platform also took measures to ensure the data collection quality, such as conducting screenings based on participants’ platform credit scores and historical acceptance rates, prohibiting repeated answering, and enabling intelligent verification to ensure the authenticity of responses.”
Finally, we also accounted for social desirability bias, which is another form of response bias. We have demonstrated it in the measurement model section [page: 14, line: 664-669]:
“Social desirability bias (SDB) is a form of response bias. In survey research, there is a risk of respondents providing socially desirable responses, potentially distorting the true relationships between variables [9,12]. To assess the presence of SDB, we administered a 13-item scale [13] and performed a correlation analysis between all variables and SDB scores. The results revealed that only one correlation exceeded the 0.40 threshold (r = -0.444), suggesting a modest influence of SDB on the participants’ responses.”
References:
- Galarraga, L.; Noriega, C.; Pérez-Rojo, G.; López, J. Emotional Competences as Predictors of Psychological Wellbeing and Quality of Life of Supplementary Grandparents Caregivers. Front. Psychol. 2024, 15, 1411634.
- Chuang, K.Y.; Yang, C.C. Interaction Patterns of Nurturant Support Exchanged in Online Health Social Networking. J. Med. Internet Res. 2012, 14, e54.
- Klein, J.; Moon, Y.; Picard, R. This Computer Responds to User Frustration: Theory, Design, and Results. Interact. Comput. 2002, 14, 119–140.
- Brock, R.L.; Lawrence, E. Too Much of a Good Thing: Underprovision versus Overprovision of Partner Support. J. Fam. Psychol. 2009, 23, 181–192.
- Armstrong, J.S.; Overton, T.S. Estimating Nonresponse Bias in Mail Surveys. J. Mark. Res. 1977, 14, 396–402.
- QuestMobile. QuestMobile 2022 Z Generation Insight Report Available online: https://www.questmobile.com.cn/research/report/316 (accessed on 9 June 2023).
- QuestMobile. QuestMobile 2022 Sportswear Market Research Report Available online: https://www.questmobile.com.cn/research/report/1595627181266014209 (accessed on 9 June 2023).
- Brandner, J.; Hood, J.C. Response Bias. In Encyclopedia of Evolutionary Psychological Science; Shackelford, T.K., Weekes-Shackelford, V.A., Eds.; Springer International Publishing: Cham, 2021; pp. 6658–6659.
- Podsakoff, P.; MacKenzie, S.; Lee, J.; Podsakoff, N. Common Method Biases in Behavioral Research: A Critical Review of the Literature and Recommended Remedies. J. Appl. Psychol. 2003, 88, 879–903.
- Lowry, P.; D’Arcy, J.; Hammer, B.; Moody, G. “Cargo Cult” Science in Traditional Organization and Information Systems Survey Research: A Case for Using Nontraditional Methods of Data Collection, Including Mechanical Turk and Online Panels. J. Strateg. Inf. Syst. 2016, 25, 232–240.
- Steelman, Z.R.; Hammer, B.I.; Limayem, M. Data Collection in the Digital Age: Innovative Alternatives to Student Samples. MIS Q. 2014, 38, 355-378.
- Ganster, D.C.; Hennessey, H.W.; Luthans, F. Social Desirability Response Effects: Three Alternative Models. Acad. Manage. J. 1983, 26, 321–331.
- Reynolds, W.M. Development of Reliable and Valid Short Forms of the Marlowe‐Crowne Social Desirability Scale. J. Clin. Psychol. 1982, 38, 119–125.
Reviewer 2 Report
Comments and Suggestions for Authors
Abstract
- Please sort keywords alphabetically
Introduction
- Why are the objectives of this study slightly different in the abstract and the introduction? Please review the terms and use of words to describe the objectives of this study
Method
- How were the population and participants selected in this study?
- What sampling technique was used in this study?
- Are there any inclusion and exclusion criteria to determine study participants?
- In table 2, why is the percentage of "Membership" not 100%? Member and non-member = 61.2% + 27.1%=....?
Results
- In Table 8, why is the mean in Tested Path: H1b and H1c negative (-)? Is there a special interpretation regarding this?
- The need for consistency in writing between p value and p-value in the manuscript. For example, in table 10 with table A1
Discussion
- Please reduce the use of "second, third" repeatedly at the beginning of paragraphs in the discussion. Please paraphrase well so that the discussion is more interesting to read and not too "stiff"
References
- Use the latest references, especially references number 63, 66, 81, 91 etc.
- Line 757: is this reference number 9? Please check again
Comments on the Quality of English LanguageThe English could be improved to more clearly express the research
Author Response
Comment1:
Introduction- Why are the objectives of this study slightly different in the abstract and the introduction? Please review the terms and use of words to describe the objectives of this study
Response1:
Thank you for your thoughtful feedback. We have reviewed the abstract and the introduction and acknowledge that there is slight inconsistency in the way the objectives of the study are presented. To improve clarity and consistency, we have revised both sections to ensure that the objectives are described uniformly as follows [page: 1, 2, line: 10-13, 123-124 ]:
“Background/Objectives: As fitness apps increasingly incorporate social interaction features, users may find themselves overwhelmed by an excess of received support, struggling to effectively manage it. Highlighting a novel recipient-centric perspective, we aim to investigate the impact of social support overload on users’ life burnout and discontinuance within fitness apps.”
“How does social support overload influence users’ life burnout and discontinuance within fitness apps?”
Comment2:
Method- How were the population and participants selected in this study?
Response2:
Thanks for your comment. The participants were recruited from Credamo, a professional survey platform with a large, diverse online sample pool of over 3 million respondents. Our survey was distributed to respondents who met the criteria of using fitness apps. The respondents were selected randomly from the platform’s sample pool, ensuring the representation of fitness app users in some degree.
Comment3:
- What sampling technique was used in this study?
Response3:
Thank you for your thoughtful feedback. We employed random sampling for participant selection. The survey was posted to Credamo’s sample pool, and the platform randomly assigned the survey to users who met the general demographic and behavioral criteria related to fitness app usage. This approach helped mitigate bias and ensured the sample was diverse and representative of the general population of fitness app users. We have revised the data collection section as follows [page: 8, line: 467-470]:
“The data collection was conducted through random sampling, with the online questionnaire being distributed to the platform’s own sample pool. The platform randomly assigned the survey to different participants, ensuring the randomness of the data.”
Comment4:
- Are there any inclusion and exclusion criteria to determine study participants?
Response4:
Thank you for your insightful question. Yes, we did implement both inclusion and exclusion criteria. Only individuals who were currently using fitness apps were included in the study. We have also clearly outlined these criteria in the data collection section to ensure transparency and to further reinforce the inclusion and exclusion criteria [page: 8, line: 459-465]:
“Given that China accounts for the largest share of global revenue in the fitness app market [1], we decided to collect data from Chinese respondents. The inclusion criteria comprised the following: (a) current users of fitness apps; (b) those who had the ability to read and write; (c) those who consented to participate in the study. There were also exclusion criteria, including the following: (a) those who were not registered users of fitness apps; (b) those who failed in our screening process; (c) those who were unwilling to consent to this study. ”
Comment5:
- In table 2, why is the percentage of "Membership" not 100%? Member and non-member = 61.2% + 27.1%=....?
Response5:
Thank you for your insightful feedback. We apologize for the oversight in Table 2 regarding the percentage of “Membership.” It was indeed a typographical error. The correct percentages are 72.9% for members and 27.1% for non-members. This has now been updated in the table, and the revised information can be found in Table 1 of the manuscript [page: 9, line: 507]. Thank you for pointing this out.
Comment6:
Results- In Table 8, why is the mean in Tested Path: H1b and H1c negative (-)? Is there a special interpretation regarding this?
Response6:
Thank you for this comment. “Mean” in Table 8 represents the sample mean of path coefficient. It reflects the average direction and magnitude of the relationships between emotional support overload and both relatedness need frustration (H1b) and competence need frustration (H1c) across the multiple resampled datasets generated during the bootstrapping procedure. We apologize for the unclear expression of “mean” in the table header. The table header has now been updated to “Path Coefficient (Sample Mean)” for better clarity [page: 15, line: 711].
Comment7:
- The need for consistency in writing between p value and p-value in the manuscript. For example, in table 10 with table A1
Response7:
Thank you for your helpful feedback regarding the consistency in writing. We have now revised the manuscript and standardized the term “p-value” throughout.
Comment8:
Discussion- Please reduce the use of "second, third" repeatedly at the beginning of paragraphs in the discussion. Please paraphrase well so that the discussion is more interesting to read and not too "stiff"
Response8:
Thank you for your valuable suggestion. We have incorporated alternative transitional words and phrases to ensure a smoother flow while maintaining logical coherence throughout the discussion section.
Comment9:
References- Use the latest references, especially references number 63, 66, 81, 91 etc.
Response9:
Thank you for your suggestion regarding the references. We have thoroughly reviewed the reference list and updated most of the citations to include more recent literature. However, we have kept some of the original theoretical articles and method guidelines, as they are foundational to the research and essential for the framework of our study.
Comment10:
- Line 757: is this reference number 9? Please check again
Response10:
Thank you for pointing this out. We have double-checked the reference and confirm that the citation in line 757 is indeed incorrect. We have corrected all the citations to ensure they accurately match the corresponding reference in the reference list.
Comment11:
The English could be improved to more clearly express the research.
Response11:
Thanks for your comments! We have used professional language editing offered by MDPI Author Services and hope the writing quality of our revised manuscript can be enhanced.
References
- Statista. Fitness Apps - Worldwide Available online: https://www.statista.com/outlook/hmo/digital-health/digital-fitness-well-being/digital-fitness-well-being-apps/fitness-apps/worldwide (accessed on 24 April 2023).
Reviewer 3 Report
Comments and Suggestions for Authors
Dear authors,
It was my pleasure to read/review this manuscript. As an active user of fitness apps, and someone who has dealt with some level of burnout with them, I find this topic to be crucial to the user experiences with these apps. This work addresses an important gap, social support overload and burnout, that has not previously been studied sufficiently. I am excited to provide my comments that hopefully will increase readership and citation of this invaluable work, which is sure to have far-reaching implications for the developers of fitness apps.
Abstract:
18. “Results highlight the significant mediating role of need frustration”. I don’t know what this means from the abstract alone.
Introduction:
84. An additional behavioral outcome measure would be beneficial here: app-use time/frequency. Total discontinuation is of course the ultimate behavioral outcome, but it would help to know if participants slowly burnout from these apps, or if social support overload leads users to immediately drop the apps altogether.
302. There appears to be a typo here: “Relatedness need get frustrated”
318. There is a similar typo here
Methods:
337. Data collection should really come before instrument selection.
Discussion:
498. These insignificant findings point to a need to include measures of these forms of social support, autonomy, and self-efficacy. It could be that the receipt of ample emotional support overrides the impacts of emotional support overload.
563. How would app managers adjust settings on workouts? The majority of these apps are trackers, rather than apps that provide the user with specific workouts. These intensity levels and workout durations are often set by the user. This seems like it would be exceedingly difficult for app developers to incorporate.
565. Might some of the suggestions in this paragraph further increase support overload, rather than foster competence? For example, leaderboards and challenges are increasing the amount of support, and social comparison, that these users experience.
576-578. In intervention studies that form peer groups, grouping peers who are non-proficient or have low levels of self-efficacy with a platform often results in negative social support, as these users provide each other with deleterious advice, rather than supporting each other. Could the authors expand upon how grouping these users together would address relatedness need frustration without increasing the amount of engagement in false information?
595. My mention of inclusion of measures of social support, autonomy, and self-efficacy could be mentioned here. I understand that it may overcomplicate the model presented here to include these, but it is a limitation to not investigate how these factors might interact/explain some of the impacts of overload on burnout/discontinuation.
Author Response
Comment1:
18. “Results highlight the significant mediating role of need frustration”. I don’t know what this means from the abstract alone.
Response1:
Thank you for your feedback. We have revised the wording to clarify the meaning. The revised abstract is as follows [page: 1, line: 14-24]:
“Methods: Utilizing Social Support Theory and Basic Psychological Needs Theory, we develop a model to examine how emotional, network, and informational support overload affect life burnout and discontinuance through the frustration of basic psychological needs: autonomy, competence, and relatedness. A total of 443 fitness app users were included in our study, and we employed Structural Equation Modeling (SEM) to empirically test this model. Results: The results highlight the significant mediating role of the frustration of basic psychological needs between social support overload and life burnout/discontinuance. Network and informational support overload positively correlate with frustration of all needs, whereas emotional support overload shows a complex relationship. All need frustrations are linked to life burnout, but only autonomy and relatedness frustrations significantly lead to discontinuance. Additionally, gender and app use proficiency are significant control variables impacting discontinuance.”
Comment2:
84. An additional behavioral outcome measure would be beneficial here: app-use time/frequency. Total discontinuation is of course the ultimate behavioral outcome, but it would help to know if participants slowly burnout from these apps, or if social support overload leads users to immediately drop the apps altogether.
Response2:
Thank you for your insightful suggestion. App-use time and frequency were treated as control variables in our study, and our analysis showed that they did not significantly impact the final outcomes. We also acknowledge the possibility that the relationships between the variables may evolve over time, which might have implications for long-term usage patterns. To address this, we have added this consideration as a limitation in the discussion section as follows [page: 20, line: 908-915]:
“Additionally, while total discontinuance serves as the ultimate behavioral outcome, the process leading to discontinuance may vary among users. It would be valuable to explore whether users experience a gradual burnout or whether the overload of social support leads to immediate abandonment of the app. This distinction could provide deeper insights into user behavior over time. Future research could benefit from adopting longitudinal or time-lagged study designs to track changes in app use over extended periods, capturing the dynamic nature of user engagement and the gradual impact of factors such as social support overload.”
Comment3:
302. There appears to be a typo here: “Relatedness need get frustrated”. 318. There is a similar typo here
Response3:
Thank you for your feedback. The typo in line 302 has been corrected to “When users’ relatedness needs are frustrated. [page: 7, line: 407]” The typo in the title at line 318 has now been corrected to “Competence Need Frustration, Life Burnout, and Discontinuance. [page: 7, line: 422]” We apologize for any confusion caused.
Comment4:
337. Data collection should really come before instrument selection.
Response4:
Thank you for your feedback. We have revised the manuscript to present the data collection section before the instrument selection.
Comment5:
498. These insignificant findings point to a need to include measures of these forms of social support, autonomy, and self-efficacy. It could be that the receipt of ample emotional support overrides the impacts of emotional support overload.
Response5:
Thank you for your suggestion. We acknowledge that it is indeed possible that the receipt of ample emotional support could override the impacts of emotional support overload. To address this, we have elaborated on this potential explanation in the discussion section [page: 18, line: 774-784]:
“Regarding the insignificant relationship between emotional support overload and autonomy need frustration, it suggests that excessive emotional support does not contribute to feelings of autonomy frustration. The possible reason for this result could be attributed to the characteristics of the main user group of fitness apps. The main users of fitness apps are young and middle-aged users who possess strong self-regulation and coping skills, enabling them to effectively handle emotional support overload [1] and then mitigate the autonomy need frustration. Furthermore, it is possible that the positive effects of receiving ample emotional support, such as a sense of validation and encouragement [2], may outweigh the negative impacts of emotional support overload. As a result, the overall emotional support may help users feel more empowered, counteracting any potential frustration related to autonomy.”
Additionally, regarding the absence of other measures, we have included this point in the limitations section of the manuscript [page: 20, line: 916-925]:
“Finally, it is important to explore additional mechanisms underlying the effects of social support overload on the negative psychological and behavioral outcomes of fitness app users. Although our mediating analysis reveals the role of BPN frustration, it is worth noting that some focal relationships are partially mediated by BPNs. Future studies can also investigate how factors such as social support, autonomy, and self-efficacy might interact with these negative factors, offering further insights into how these factors influence the experience of overload. Therefore, other potential mechanisms can be considered based on valid theoretical perspectives in future studies. Additionally, deeper investigations are warranted to develop and evaluate app design strategies that effectively tackle social support overload in fitness apps.”
Comment6:
563. How would app managers adjust settings on workouts? The majority of these apps are trackers, rather than apps that provide the user with specific workouts. These intensity levels and workout durations are often set by the user. This seems like it would be exceedingly difficult for app developers to incorporate.
Response6:
Thank you for your insightful comment. We believe that this challenge can be addressed using advanced technologies. Specifically, AI-driven algorithms and backend data analysis can help provide personalized workout adjustments based on users’ activity data. By analyzing users’ past performance, preferences, and progress, these systems can suggest optimal workout settings. In fact, there are many fitness apps embedded with AI such as FitnessAI and Fitbod to optimize training and offer guidance automatically. We have further elaborated on this point in the practical implications of discussion section [page: 19, line: 859-864]:
“In addition, given the mediating role of BPNs frustration, interventions can be developed to target its reduction. For autonomy need frustration, fitness app managers can provide users with adjustable settings, such as those for workout duration, intensity levels, and rest intervals, to grant users greater control over their fitness routines. Advanced technologies such as AI-driven algorithms and backend data analysis can be employed to make personal recommendations based on users’ activity data.”
Comment7:
565. Might some of the suggestions in this paragraph further increase support overload, rather than foster competence? For example, leaderboards and challenges are increasing the amount of support, and social comparison, that these users experience.
Response7:
Thank you for your suggestion. We acknowledge that there is indeed a possibility that some of the suggestions, such as leaderboards and challenges, could inadvertently increase support overload by heightening social comparison among users. While we emphasize the importance of maintaining a balance in the design of these elements. Specifically, we recommend incorporating gradual, progressive challenges that allow users to increase difficulty based on their individual levels, ensuring that they always feel a sense of control over their fitness journey. This approach helps to reduce unnecessary pressure and fosters a positive experience, rather than overwhelming users. We have further refined and strengthened this logic in the discussion section to clarify this point as follows [page: 19, line: 864-872]:
“For the frustration of relatedness and competence needs, incorporating gamification elements would be effective. Implementing leaderboards or team-based challenges in fitness apps can foster friendly competition and a sense of belonging among users. Meanwhile, participation in leaderboards and challenges can be made optional, giving users the freedom to engage with these features at their discretion. Virtual badges or points can be used as rewards to reinforce users’ sense of competence when they achieve milestones in their fitness journey. Furthermore, offering challenging yet achievable exercise activities that gradually increase in difficulty allows users to challenge themselves while still feeling accomplished.”
Comment8:
576-578. In intervention studies that form peer groups, grouping peers who are non-proficient or have low levels of self-efficacy with a platform often results in negative social support, as these users provide each other with deleterious advice, rather than supporting each other. Could the authors expand upon how grouping these users together would address relatedness need frustration without increasing the amount of engagement in false information?
Response8:
Thank you for your suggestion. To address this concern, we propose two measures to ensure that peer group interactions are constructive and beneficial. First, fitness app managers could establish and enforce group rules that emphasize positive, supportive, and accurate communication. Clear guidelines, along with the promotion of a healthy community culture, can mitigate the risk of users sharing deleterious advice. Second, a feedback mechanism could be introduced, allowing users to evaluate the quality of their interactions within the peer groups. This feedback could be used to refine grouping algorithms and ensure that users are paired with peers who positively contribute to their experience. We have incorporated these considerations into the revised manuscript to address this issue [page: 19, line: 873-885]:
“Lastly, recognizing the impact of gender and proficiency of use, interventions can be tailored to mitigate discontinuance rates in fitness app usage. Building upon the stronger positive association between relatedness need frustration and discontinuance among female and nonproficient users, fitness app managers can make use of their database of registered users to recommend compatible peers with similar backgrounds to women and nonproficient users. Fitness app managers can implement measures to guide the nature of interactions. This includes establishing clear group rules that promote supportive, accurate, and constructive communication, as well as fostering a positive community culture. Additionally, a feedback mechanism could allow users to evaluate the quality of their social interactions, enabling managers to refine grouping algorithms and better tailor peer recommendations. By facilitating the establishment of social relationships among these user groups, fitness app managers can effectively address users’ relatedness need frustration and enhance their engagement with fitness apps.”
Comment9:
595. My mention of inclusion of measures of social support, autonomy, and self-efficacy could be mentioned here. I understand that it may overcomplicate the model presented here to include these, but it is a limitation to not investigate how these factors might interact/explain some of the impacts of overload on burnout/discontinuation.
Response9:
Thank you for your valuable suggestion. We agree that the inclusion of factors such as social support, autonomy, and self-efficacy could provide deeper insights into the mechanisms underlying social support overload and its impacts on burnout and discontinuance. We have revised the limitation section as follows [page: 20, line: 916-925]:
“Finally, it is important to explore additional mechanisms underlying the effects of social support overload on the negative psychological and behavioral outcomes of fitness app users. Although our mediating analysis reveals the role of BPN frustration, it is worth noting that some focal relationships are partially mediated by BPNs. Future studies can also investigate how factors such as social support, autonomy, and self-efficacy might interact with these negative factors, offering further insights into how these factors influence the experience of overload. Therefore, other potential mechanisms can be considered based on valid theoretical perspectives in future studies. Additionally, deeper investigations are warranted to develop and evaluate app design strategies that effectively tackle social support overload in fitness apps.”
References
- Galarraga, L.; Noriega, C.; Pérez-Rojo, G.; López, J. Emotional Competences as Predictors of Psychological Wellbeing and Quality of Life of Supplementary Grandparents Caregivers. Front. Psychol. 2024, 15, 1411634.
- Chuang, K.Y.; Yang, C.C. Interaction Patterns of Nurturant Support Exchanged in Online Health Social Networking. J. Med. Internet Res. 2012, 14, e54.